# A Study on Chloride Corrosion Resistance of Reactive Powder Concrete (RPC) with Copper Slag Replacing Quartz Sand under Freeze–Thaw Conditions

**DOI:** 10.3390/ma17010212

**Published:** 2023-12-30

**Authors:** Jun Li, Xianzhang Liu, Minghao Chen, Lijun Tian, Jiao Liu

**Affiliations:** 1School of Civil Engineering, Liaoning Technical University, Fuxin 123000, China; lj0801@126.com (J.L.);; 2Liaoning Key Laboratory of Mine Subsidence Disaster Prevention and Control, Liaoning Technical University, Fuxin 123000, China

**Keywords:** copper slag, RPC, freeze–thaw cycles, chloride ion diffusion

## Abstract

In order to study the influence of freeze–thaw cycles on chloride ion corrosion resistance of RPC with copper slag (CS) instead of quartz sand (QS), the 28d uniaxial compressive strength (UCS) of CSRPC with a different CS substitution rate was investigated by unconfined compression tests. The electric flux test method was used to study the chloride ion diffusion resistance of CSRPC after freeze–thaw cycles, and the pore size distribution was obtained through the nuclear magnetic resonance (NMR) method. Then, a mathematical relationship between the chloride ion diffusion coefficient and the pore fractal characteristic parameter T was established to study the effect of freeze–thaw cycles on chloride ion diffusion. Finally, SEM/EDS, XRD, and DTG methods were combined to study the influence of the distribution of Friedel’s salts generated after freeze–thaw cycles on chloride ion diffusion in CSRPC. The results indicate that CS has a micro aggregate effect and pozzolanic activity, which can effectively improve the chloride ion diffusion resistance of CSRPC after freeze–thaw cycles. In addition, the electric flux of CSRPC decreases with the increase in freeze–thaw cycles, and the chloride diffusion coefficient is closely related to the pore fractal dimension.

## 1. Introduction

The long-term natural stockpiling of CS not only occupies a lot of land resources but also leads to air dust pollution. The resource utilization and safe disposal of tailings generated by metal smelting are increasingly attracting attention from countries around the world. Copper slag (CS) is a solid waste produced in the process of pyrometallurgical copper smelting. The production of 1 t of metal copper will be accompanied by 2.2 tons of CS [1], which contains a large amount of iron and cannot be enriched again through beneficiation. The traditional treatment method is to grind CS as an iron correction material, but due to the high carbon emissions and high energy consumption in cement production, it is also difficult for cement companies to absorb CS on a large scale in a short period of time. Therefore, it is necessary to develop and utilize solid waste CS rationally. Because CS contains a large amount of silica, it is possible to replace quartz sand with high-performance reactive powder concrete (RPC).

CS contains a large amount of iron (up to 35% to 40%), as well as small amounts of copper and zinc [2], which can be recovered through mature processes, such as thermal reduction and thermal decomposition [3,4,5,6]. After deep reduction and extraction of zinc and iron, the main crystalline phase of CS is belite (C_2_S). After quenching and tempering treatment, part of C_2_S is converted into alite (C_3_S), which is basically ordinary Portland clinker with strength close to 42.5 ordinary Portland cement, indicating that copper slag has good potential in preparing ordinary cement [7,8,9,10,11,12]. The incorporation of CS can effectively improve the water absorption of concrete, and the clinker produced with copper slag poses few environmental problems. Zong et al. [13] found that the durability of CS concrete is better than ordinary silicate concrete CS instead of fine aggregate. Zhang et al. [14] discovered that the early strength of concrete increased rapidly, and the compressive bending resistance became better when the river sand was replaced by CS. Ambley et al. [15] validated the feasibility of using CS as fine aggregate in ultra-high-performance concrete (UHPC) and produced ultra-high-performance concrete with a compressive strength greater than 150 MPa.

In 1993, Richard and Cheyrezy of Bouygues Co. in France replicated the high-density cement uniform system and discovered that by eliminating the coarse aggregate within this system, RPC could exhibit exceptional mechanical properties. [16]. They used quartz sand (QS) with a maximum particle size of 400~600 μm as aggregate mixed with an appropriate number of short fibers and reactive mineral admixtures, such as silica fume, and finally obtained cement-based composite materials with high strength, high toughness, and excellent mechanical properties. This kind of concrete is called reactive powder concrete (RPC) because it improves the fineness and reactivity of the components. But, its disadvantage is that the cost is high, and it can only be used for some special projects.

In recent years, research on RPC fine aggregates has focused on the use of finely ground construction solid wastes, broken rubber tires, waste brick powder, blast furnace slag, rice husk ash, and fly ash to replace the fine aggregate or cementitious materials in the RPC formula [17,18,19,20], which has better environmental significance and reduces the project cost. Tian et al. [21] used a dolomite powder as fine aggregate and used low-energy ohmic heating (LEOH) curing instead of traditional high-temperature steam (HTS) curing to shorten the curing time to less than 3 h and reduce the direct cost by 61.2%. Li et al. [22] utilized titanium slag as fine aggregate to replace QS powder in order to produce titanium slag-based RPC that exhibits superior performance. This finding indicates that metal slag has the potential to produce RPC concrete with exceptional performance. Xue et al. [23] used steel slag as fine aggregate to improve the compressive strength of concrete specimens, enhance the hydration products of the cement matrix, and reduce the number of micro-cracks. The results showed that the surface fractal dimension has a good correlation with pore structure parameters.

Sun et al. [24] found that the resistance of cement-based materials to chloride entry mainly depends on the cumulative width of all cracks and revealed the influence of crack width and dry–wet cycles on the mineral phase, porosity, and pore size distribution near the crack. Zhu et al. [25] found that the pore size distribution of recycled aggregate concrete under chloride ion corrosion has similar fractal characteristics, and the fractal dimension increases first and then decreases with the increase in pore size. Guo et al. [26] studied the effect of construction waste composite powder materials as supplementary cementitious materials on the performance of concrete and found that the pore size fractal dimension can characterize the pore structure of concrete, which has a significant correlation with compressive strength and the chloride ion diffusion coefficient. Jin et al. [27] thought that the fractal dimension could quantitatively characterize pore structure and established a freeze–thaw damage model with fractal dimension.

CSRPC prepared from smelting waste residue CS shows potential in the hydraulic structure of the seasonal freezing zone, but the chloride diffusion mechanism of CSRPC after freeze–thaw cycles has not been studied deeply. Therefore, in this paper, the influence of the CS substitution rate on the 28d UCS of CSRPC after freeze–thaw cycles is studied through laboratory experiments. The T_2_ spectrum curve of CSRPC was tested by the NMR method to determine the pore distribution of micro-pores, meso-pores, and macro-pores, and the mathematical relationship between the chloride ion permeability coefficient and pore fractal characteristic parameter was established. Finally, the mechanism of resistance to chloride ion diffusion under the influence of freeze–thaw cycles on CSRPC is revealed by SEM/EDS, XRD, and DTG methods.

## 2. Materials and Method

### 2.1. Copper Slag (CS)

#### 2.1.1. Physical Properties of CS

As shown in Figure 1a, the CS used for the test is converter slag from Chifeng Fubang (for China) Copper Co., Ltd., Inner Mongolia, China, which is a kind of tailing slag made by crushing converter copper slag to form copper slag fragments with a size less than 200 mm. The ingredients are poured into an intermediate frequency furnace for stratified melting, and then they are discharged and cooled. The basic physical properties of the copper slag are shown in Table 1.

Jaw crushers are utilized for the crushing of coarse copper slag, while high-pressure roll mills are employed for the fine grinding of copper slag, ultimately yielding raw material CS. Figure 1b illustrates the particle size distribution curves of quartz sand (QS) and copper slag (CS).

#### 2.1.2. Chemical Components

The chemical components of CS are determined by XRD, which provided the basis for the preparation of CS-based RPC. The XRD results are shown in Figure 1c, and the chemical components analysis results of CS are shown in Table 2.

As can be seen in Table 2, the main chemical components of copper slag are Fe_2_O_3_, CaO, SiO_2_, and Al_2_O_3_. It is preliminarily inferred that CS may have pozzolanic activity in addition to a micro-aggregate effect.

### 2.2. Other Raw Materials

#### 2.2.1. Portland Cement

The cement used in the experiment is Longshoushan 42.5 grade Portland cement from Tieling City, Liaoning Province, China. Its basic chemical composition and physical properties are shown in Table 3.

#### 2.2.2. Silica Fume (SF)

In this experiment, a kind of ultrafine silica powder material was selected. The ultrafine silica powder material was formed by the rapid reaction and condensation of silicon gas generated at a high temperature of more than 2000 °C with oxygen in the air. Its chemical components and particle size distribution are listed in Table 4 and Table 5, respectively.

#### 2.2.3. Steel Fiber

The copper-plated straight steel fiber produced in the iron and steel plant in Anshan City, Liaoning province is selected. It has a diameter of 0.21 mm, a length of 13 mm, and a tensile strength of 3000 MPa, approximately.

#### 2.2.4. Polycarboxylate Superplasticizer (PS)

The PS used in the test is a yellow transparent liquid with a solid content of 26% and a water reduction rate greater than 30%.

### 2.3. Experimental Scheme

#### 2.3.1. Specimen Preparation

The Portland cement and silica fume are regarded as cementitious materials, and CS used to substitute for quartz sand is regarded as aggregate, and the water binder ratio W/B is controlled at 0.21 [28]. A horizontal cement mixer (Figure 2b) is selected to mix CSRPC mortar. The mass mixing ratio of each component of RPC is m(Portland cement): m(QS + CS): m(cement): m(steel fiber): m(silica fume) = 26%: 11.48%: 5.43%: 6.25%: 31%. First, the Portland cement and silica fume were put into the horizontal cement mixer for dry stirring, and 80% PS was poured into the blender after fully mixing. Then, the steel fiber, CS + QS mixture, and the remaining 20% PS were poured into the mixture to prepare fresh RPC, which was transferred to the steel mold in three layers and compacted with a vibrating table. The size of the vibrating table is 2 m × 1 m, and its load capacity, frequency, and amplitude are 1500 kg, 66 Hz, and 8 cm, respectively. Finally, the CSRPC samples were released after being held at room temperature for 24 h and cured under standard curing conditions for 28 days.

#### 2.3.2. The 28d UCS Test

In order to study the influence of the CS substitution rate and freeze–thaw cycles on the 28d UCS and electric flux of CSRPC, the experiment scheme of the CS substitution rate and freeze–thaw cycles are shown in Table 6. According to the test scheme, the CSRPC specimens (3 in each group) with 150 mm × 150 mm × 150 mm are prepared. As shown in Figure 2c, the YE-2000 pressure tester is used to test the 28d UCS of CSRPC.

#### 2.3.3. Rapid Freeze–Thaw Test and NMR Test

As shown in Figure 2d, the GDR3-9 rapid freeze–thaw tester is used to conduct freeze–thaw experiments on CSRPC, and the freeze–thaw cycles are set as 50, 100, 150, 200, and 250, respectively. The freezing temperature is maintained at −20 °C, and the melting temperature is maintained at 20 °C. After the freeze–thaw test, the MesoMR23-060H-I nuclear magnetic resonance (NMR) tester is used to obtain the T_2_ spectral curves of CSRPC with different substitution rates so as to determine the pore diameter of CSRPC after freeze–thaw. The distribution of the pore diameter of CSRPC is determined by the relationship between relaxation time and intensity on the T_2_ spectral curve.

#### 2.3.4. Electric Flux Test

According to the Chinese standard (JTJ 275-2000), the chloride ion diffusion characteristics of CSRPC were tested by the electric flux method [29]. A cylindrical CSRPC with a height of 51 mm and a diameter of 95 mm was prepared and cured in a saturated calcium hydroxide (Ca(OH)) solution for 28 days. As shown in Figure 2d, the DTL-6 electric flux meter was used to measure the electric flux. The sodium chloride (NaCl) solution with a concentration of 3.0% and the sodium hydroxide (Na(OH)) solution with a concentration of 0.3 mol/L were injected into the test tank connected to the negative and positive electrodes, respectively. A 60 V DC voltage was applied, and the test tank was kept full of solution. When the current value changed significantly, the current value was recorded every 10 min, and when the current changed little, the current value was recorded every 30 min until the current was on for 6 h. The electric flux *Q* of 6 h can be obtained by integrating the current-time curve.

#### 2.3.5. SEM, XRD, and TG Analysis

As shown in Figure 2e, the SEM/EDS method was used to analyze the geometric characteristics of pores under the influence of freeze–thaw cycles and the microscopic characteristics of Friedel’s salts produced under the diffusion of chloride ions. The compound composition of CSRPC after chloride ion diffusion was analyzed by XRD, and the content of each chemical component was determined by thermogravimetric (TG) analysis.

## 3. Results and Discussion

### 3.1. Effect of the CS Substitution Rate on the Mechanical Properties of CSRPC

Figure 3a,b show the test results of the 28d UCS and bulk density of CSRPC specimens with different CS substitution rates before and after freeze–thaw cycles.

As can be seen in Figure 3a, before freeze–thaw cycles, when the CS substitution rate is lower than 70%, the 28d UCS and bulk density of group A increase with the increase in the CS substitution rate. However, when the CS substitution rate exceeds 70%, the 28d UCS and bulk density decrease with the increase in the CS substitution rate. After 50 freeze–thaw cycles, the 28d UCS and bulk density of group B specimens increase with the increase in the CS substitution rate. This is because before the freeze–thaw cycle, the internal structure of CSRPC is relatively dense. With the increasing CS substitution rate, the active silica and alumina in CS participate in alkali-activated reactions with CH, leading to the formation of C-S-H gel. Specifically, in an alkaline environment, alumina reacts with reactive siliceous materials to produce hydrated aluminosilicates, which, in turn, promotes the formation of C-S-H gel. The C-S-H gel plays a cementing role among the granular particles, enhancing the overall strength of the sample. However, when the CS substitution rate exceeds 70%, the C-S-H gel absorbs water, undergoes volume expansion, and causes cracking, ultimately leading to a reduction in 28d UCS. After freeze–thaw cycles, the pores of CSRPC samples expand due to frost-heaving forces. The C-S-H gel, formed via alkali-bone reactions, fills these pores and cement particles, enhancing the compactness and strength of CSRPC. The results indicate that the replacement of QS with CS significantly improves the freeze–thaw damage resistance of RPC.

It can be seen in Figure 3b that when the CS substitution rate is 0, the 28d UCS in group C after 100, 150, 200, and 250 freeze–thaw cycles decreased by 3.58%, 19.07%, 36.43%, and 56.34%, respectively, and the bulk density decreased by 3.06%, 7.38%, 19.96%, and 25.43%, respectively, compared with before the freeze–thaw cycles (A1). When the CS substitution rate was 100%, after 100, 150, 200, 250 freeze–thaw cycles in group D, the 28d UCS decreased by 7.6%, 16.55%, 22.96%, and 36.76%, and the bulk density decreased by 5.59%, 7.79%, 13.35%, and 15.75%, respectively, compared with A4. The 28d UCS and bulk density of RPC decreased with the increase in freeze–thaw cycles, and this is because the frost-heaving force caused the expansion of the internal pore, leading to a reduction in the compactness of CSRPC after freeze–thaw cycles. However, the 28d UCS and bulk density of CSRPC with a 100% substitution rate are higher than CSRPC with a low substitution rate. It can be seen that the decrease in bulk density caused by the expansion of the pore structure has a significant impact on the deterioration of the mechanical properties of RPC, and CSRPC has better freeze–thaw damage resistance than QSRPC.

### 3.2. Pore Structure Characteristics of CSRPC

#### 3.2.1. CSRPC Relaxation Properties

According to the principle of nuclear magnetic resonance (NMR), the transverse relaxation time T_2_ is mainly affected by the surface fluid relaxation effect, and the relationship between T_2_ and $S/V_{p}$ can be presented as [30]:(1)$$\frac{1}{T_{2}}=\rho \left(T_{2}\right)\frac{S}{V_{p}}$$
where $\rho \left(T_{2}\right)$ is the surface relaxation intensity, $m\cdot {\mathrm{ms}}^{-1}$, depending on the pore surface properties and mineral composition, which can be approximately taken as $0.5\times 10^{-8}m\cdot {\mathrm{ms}}^{-1}$; S is the total pore surface area, $m^{2}$; and $V_{p}$ is the total pore volume, $m^{3}$, and the relationship between $S/V_{p}$ and pore size r can be expressed as:(2)$$\frac{S}{V_{p}}=\frac{F_{s}}{r}$$
where $F_{s}$ is the pore geometry factor. Since the pore channel between pores can be regarded as a cylinder, $F_{s}$ can be approximately taken as 2.

By substituting Equation (2) into Equation (1), the relationship between pore size *r* and transverse relaxation time T_2_ can be approximately expressed as:(3)$$r=10^{-8}T_{2}\left(m\right)$$

It can be seen in Equation (3) that there is a close relationship between the pore size *r* and the transverse relaxation time T_2_, so the pore size distribution of CSRPC can be obtained by the T_2_ spectrum. According to the size of the pore, the pore of CSRPC can be divided into three types: micro-pore: 10^−4^ μm ≤ *r* < 10^−2^ μm (0.01 ms ≤ T_2_ < 1 ms); meso-pore: 10^−2^ μm ≤ *r* < 1 μm (1 ms ≤ T_2_ < 10^2^ ms); and macro-pore: 1 μm ≤ *r* < 100 μm (10^2^ ≤ T_2_ < 10^4^ ms). Figure 4a–d show the T_2_ spectrum of CSRPC in groups A ~ D, respectively.

The area enclosed by the T_2_ spectral curve and the horizontal axis corresponds to the total pore volume, and the pore size corresponding to the peak position of the T_2_ spectrum curve is the most probable pore size MPPS. The larger the MPPS is, the easier chloride ions diffuse in RPC. It can be seen in Figure 4a,b that with an increase in the CS substitution rate, the MPPS gradually decreases, indicating that with an increase in the CS substitution rate, the pore connectivity of CSRPC decreases, and the chloride ion diffusion resistance is significantly enhanced. With an increase in freeze–thaw cycles, the MPPS gradually increases, indicating that freeze–thaw cycles lead to the enhancement of pore connectivity of CSRPC and the weakening of chloride ion diffusion resistance.

#### 3.2.2. CSRPC Pore Structure Characteristics

The spectrum area and peak proportions of micro-pores, meso-pores, and macro-pores of CSRPC determined according to the T_2_ spectrum in Figure 4a–d are summarized in Table 7.

By comparing group A and group B in Table 7, it can be seen that with the increase in the CS substitution rate, the proportion of meso-pores gradually increases, while the proportion of macro-pores gradually decreases, indicating that the substitution of CS for QS is beneficial to the refinement of pores in CSRPC and contributes to the transformation of meso-pores and macro-pores into micro-pores gradually. The T_2_ spectral area after 50 freeze–thaw cycles is larger than before the freeze–thaw cycles under the same CS substitution rate, indicating that the pore volume expansion under freeze–thaw cycles is the main reason for the freeze–thaw damage of CSRPC. The increasing of the CS substitution rate can effectively reduce the proportion of meso-pores and macro-pores and improve the overall integrity of RPC.

By a comparative analysis of pore size distribution in group C and group D, it can be seen that the micro-pore proportion of CSRPC with a 100% CS substitution rate is generally higher than with a low CS substitution rate. With the increase in freeze–thaw cycles, the proportion of micro-pores gradually decreases, while the proportion of meso-pores and macro-pores gradually increases, indicating that the micro-pores gradually transform into meso-pores and macro-pores after freeze–thaw cycles. The use of CS instead of QS can effectively reduce the proportion of macro-pores in CSRPC after freeze–thaw cycles and restrict the volume expansion caused by freeze–thaw damage.

#### 3.2.3. Fractal Dimension of the Pore of CSRPC

The more tortuous the pore is, the greater the fractal dimension of the pore throat is, and the greater the collision probability of chloride ions moving in the pore channel with the inner wall of the pore. This is more unfavorable for the diffusion of chloride ions. It is assumed that the pore size distribution is continuous, and the pore size corresponding to the T_2_ spectrum in $\left(T_{2}\left(i\right)-\delta /2,T_{2}\left(i\right)+\delta /2\right]$ is $r_{i}$, and the corresponding pore volume $V\left(r_{i}\right)$ can be expressed as:(4)$$V\left(r_{i}\right)=\frac{\int_{T_{2}\left(i\right)-\delta /2}^{T_{2}\left(i\right)+\delta /2}I\left[T_{2}\left(i\right)\right]dT_{2}}{\int_{T_{2,\min}}^{T_{2,\max}}I\left[T_{2}\left(i\right)\right]dT_{2}}\varphi V_{0}$$
where φ is overall porosity, $V_{0}$ is the total volume, and $I\left(T_{2}\right)$ is the intensity of the T_2_ spectrum. The whole T_2_ spectrum can be divided into 100 bands according to the relaxation time T_2_, and each band’s width is δ.
(5)$$\delta =\frac{T_{2,\max}-T_{2,\min}}{100}$$
where $T_{2,\max}$ and $T_{2,\min}$ are the maximum and minimum transverse relaxation time, respectively. The fractal dimension of the pore area and pore throat can be determined by the slope of the double logarithmic functions $\ln\left[dV\left(r\right)/dr\right]∼\ln\left(r\right)$ and $\ln\left[d^{2}V\left(r\right)/dr^{2}\right]∼\ln\left(r\right)$, respectively.
(6)$$\ln\left[\frac{dV\left(r\right)}{dr}\right]∼\left(2-D_{s}\right)\ln\left(r\right)$$
(7)$$\ln\left[\frac{d^{2}V\left(r\right)}{dr^{2}}\right]∼-D_{l}\ln\left(r\right)$$
where $D_{s}$ is the fractal dimension between 2 and 3 and $D_{l}$ is the fractal dimension of pore throat between 1 and 2. The larger the fractal dimension of the pore area is, the rougher the pore surface is, and the greater the diffusion resistance of chloride ions in the pore channel is, the more unfavorable the diffusion of chloride ions is. The larger the fractal dimension of the pore throat is, the greater the tortuosity of the RPC pore connection channel is.

In 1963, B. B. Bennet proposed the RRB model to describe the particle size distribution law [31]:(8)$$R\left(x\right)=100\exp\left[-{\left(\frac{x}{X}\right)}^{n}\right]$$

Figure 5a,b give the CSRPC pore size distribution of group C and group D of CSRPC, respectively. Obviously, Equation (8) deviates from the actual pore size distribution. Therefore, the cumulative Weibull probability density function is introduced to describe the pore size distribution.
(9)$$R\left(x\right)=1-\exp\left[-{\left(\frac{x}{X}\right)}^{n}\right]$$
where *R*(*x*) is the cumulative frequency of the proportion of pores smaller than the size of *x*, *X* is the characteristic pore size, and *n* is the uniformity coefficient of the model size distribution. The effect of fitting the pore size distribution with Equation (9) is shown in Figure 5a,b.

The characteristic parameters of the chloride pore structure in RPC can be expressed as [7]:(10)$$T=\frac{\varphi \cdot S_{0}\cdot X\cdot n}{2D_{s}^{2}\cdot D_{l}}$$

The pore shape correction coefficient *S*_0_ is introduced to represent the volume deviation between the actual pore volume and the imaginary volume of the same volume sphere, which can be calculated with Equation (11).
(11)$$S_{0}=\frac{3}{4\pi }\sum_{i=1}^{n}V\left(r_{i}\right)/\sum_{i=1}^{n}r_{i}^{3}$$
(12)$$V\left(r_{i}\right)=\frac{\int_{T_{2}\left(i\right)-\delta /2}^{T_{2}\left(i\right)+\delta /2}I\left[T_{2}\left(i\right)\right]dT_{2}}{\int_{T_{2,\min}}^{T_{2,\max}}I\left[T_{2}\left(i\right)\right]dT_{2}}\varphi V_{0}$$

Considering Equations (4), (11), and (12), Equation (10) can be transformed into:(13)$$T=\frac{3\varphi^{2}XnV_{0}}{8\pi D_{s}^{2}D_{l}A_{t}}\sum_{i=1}^{n}\frac{\int_{T_{2}\left(i\right)-\delta /2}^{T_{2}\left(i\right)+\delta /2}I\left[T_{2}\left(i\right)\right]dT_{2}}{\int_{T_{2,\min}}^{T_{2,\max}}I\left[T_{2}\left(i\right)\right]dT_{2}}/\sum_{i=1}^{n}r_{i}^{3}$$

Pores with sizes of 0.01~2 µm are used to calculate the fractal dimension *D_l_* of the pore throat, and the pores with sizes of 2~1000 µm are used to calculate the fractal dimension *D_S_* of the pore surface [26].

Figure 6a–d show the fractal dimension fitting curves of QSRPC in group C and CSRPC in group D. Table 8 shows the calculation results of the fractal dimensions *D_l_* and *D_S_* and fractal characteristic parameter *T*.

### 3.3. Effect of Freeze–Thaw Cycles on Electric Flux

As the electric conductivity changes with time, electric flux is used to represent the chloride ion diffusion characteristics. Figure 7a,b show the electric flux and fractal dimension of QSRPC in Group C and CSRPC in Group D after 6 h of DC power on, respectively.

It can be seen in Figure 7a,b that the electric flux of CSRPC increases gradually with the increase in freeze–thaw cycles. The 6 h electric flux of CSRPC with a 100% CS substitution rate is relatively lower than QSRPC, indicating that CSRPC has better chloride diffusion resistance than QSRPC. This is because QS has relatively stable chemical properties and hardly reacts with the hydration products of Portland cement. It only plays a micro aggregate effect in RPC and has little effect on refining pore structure, while the active SiO_2_ in CS reacts with CH (Ca (OH)_2_) generated by cement hydration to generate C-S-H gel with a low Ca/Si ratio, which refines the internal pores of RPC and improves the compactness of CSRPC. Therefore, compared with QS, the addition of CS is more conducive to enhancing the chloride ion diffusion resistance of RPC.

The MPPS of QSRPC increased from 6.53 nm to 9.21 nm after 28 days, and the corresponding pore throat fractal dimension increased from 1.56 to 1.63, while the MPPS of CSRPC increased from 7.24 nm to 12.74 nm, and the pore throat fractal dimension increased from 1.47 to 1.58. It can be inferred that the pozzolanic reaction of the active SiO_2_ in the CSRPC will lead to an increase in the degree of pore throat twist, a greater resistance to chloride ion diffusion, and a significant decrease in pore connectivity. Considering the relationship between conductivity and electric flux, the conductivity of the specimen can be calculated according to Equation (14), with the accuracy of 0.01 × 10^−4^ S.
(14)$$C_{i}=\frac{1}{R_{i}}=\frac{Q}{Ut}$$
where $C_{i}$ is the electric conductivity at room temperature and $R_{i}$ is the resistance. The dimensions are S and Ω, respectively. *Q* represents the electric flux density of 6 h (C), *U* stands for the DC voltage (V), and *t* is the power-on time, which is 3600 s.

The conductivity is corrected according to Equation (15) [31].
(15)$$C_{20}=e^{2130\left(1/T_{i}-1/T_{20}\right)}C_{i}$$
where $T_{i}$ is the absolute temperature of the saturated sodium hydroxide solution (K), and $T_{20}=293K$. According to the Nernst Plank equation, the chloride ion diffusion coefficient *D* can be deduced as:(16)$$D=0.235\times 10^{-2}C_{20}$$
where *b* = 0.235 × 10^−2^ is the constant related to the concrete type. By introducing Equations (14) and (15) into Equation (16), we can obtain:(17)$$D=0.235\times 10^{-2}\frac{Q}{Ut}e^{2130\left(1/T_{i}-1/T_{20}\right)}$$

The chloride ion diffusion coefficient *D* can be calculated by Equation (17), and the relationship of the calculated chloride ion diffusion coefficient *D* and electric flux of CSRPC and QSRPC are shown in Figure 8a; *D* is significantly reduced and lower than QSRPC. Therefore, by modifying Equation (17), we can obtain:(18)$$D=0.235\times 10^{-2}a\frac{Q}{Ut}e^{2130\left(1/T_{i}-1/T_{20}\right)}+b$$
where *a* and *b* are the correction coefficients. The effect of fitting the calculated *D* with Equation (18) is shown in the fitting straight line in Figure 8a, and the fitting result is given in Equation (19).
(19)$$\left\{\begin{array}{l} D_{\mathrm{QSRPC}}=1.8359\times 10^{-11}Q-1.8368\times 10^{-8}\\ D_{\mathrm{CSRPC}}=1.5087\times 10^{-11}Q-1.4875\times 10^{-8} \end{array}\right.$$

It can be seen in Figure 8a that although the fractal dimension of the pore throat and pore area increases, which is unfavorable for chloride diffusion, the electric flux still increases, indicating that the chloride ion diffusion coefficient has no direct relationship with the fractal dimension of pores, so it is necessary to introduce the pore fractal characteristic parameter that can comprehensively describe the pore size and complexity. The fitting line in Figure 8b shows the relationship between RPC electric flux and pore fractal characteristic parameter *T*, and the fitting result is given in Equation (20).
(20)$$\left\{\begin{array}{l} D_{\mathrm{QSRPC}}=1.5308\times 10^{-8}T+2.6879\times 10^{-8}\\ D_{\mathrm{CSRPC}}=9.7025\times 10^{-9}T+2.5681\times 10^{-8} \end{array}\right.$$
where *D*_QSRPC_ and *D*_CSRPC_ are the chloride ion diffusion coefficients of QSRPC and QSRPC, respectively. As can be seen in Figure 8b, the pore fractal characteristic parameter *T* can be used to characterize the tortuosity of small pores and the roughness of large pores of RPC, and there is a corresponding relationship between the pore fractal characteristic parameter and chloride diffusion coefficient. Under the influence of freeze–thaw cycles, both *D*_QSRPC_ and *D*_CSRPC_ have a linear relationship with the pore fractal dimension characteristic parameter *T*.

### 3.4. Analysis of Micro-Structure and Chemical Composition

#### 3.4.1. Effect of Freeze–Thaw on the Geometric Characteristics of CSRPC Pores

Figure 9 and Figure 10 show the SEM, pore area distribution, and the correlation between the length of the major and minor axes of CSRPC and QSRPC, respectively, after 250 freeze–thaw cycles, and Figure 11 shows the pore roundness distribution with CS substitution rates of 0 and 100%.

IPP image processing software (Intel Parallel Studio XE) is used to extract the area and major and minor axis geometric parameters of pores on the SEM. As can be seen in Figure 9a and Figure 10a, after 250 freeze–thaw cycles, there are many micro-cracks generated by the free water frost-heaving force in RPC. The major axes of the largest micro-cracks of QSRPC and CSRPC are 13.36 μm and 8.57 μm, respectively, and the minor axes are 1.12 μm and 1.27 μm, respectively. As can be seen in Figure 9b and Figure 10b, the total pore area of QSRPC and CSRPC are 100.53 μm^2^ and 63.93 μm^2^, respectively. The average pore area is 1.68 μm^2^ and 1.16 μm^2^, respectively. As can be seen in Figure 9c and Figure 10c, the correlation coefficients of the major axis and the minor axis of QSRPC and CSRPC are 0.7713 and 0.8093, respectively.

As can be seen in Figure 11, the mean roundness of QSRPC and CSRPC are 1.41 and 1.26, respectively. Compared with QSRPC, both the pore roundness and the correlation coefficient of the major axis and minor axis after 250 freeze–thaw cycles of CSRPC are closer to 1, indicating that the frost-heaving stress is more evenly distributed around the pore of CSRPC, effectively reducing the uneven expansion of freeze–thaw cracks caused by stress concentration.

#### 3.4.2. Effect of Freeze–Thaw Cycles on Chloride Diffusion Resistance of CSRPC

Figure 12a–d show the SEM/EDS of CSRPC before and after 250 freeze–thaw cycles under chloride ion diffusion conditions.

Friedel’s salt, CH, and C_3_S may all be hexagonal, and the size of Friedel’s salt ranges from 2 μm to 3 μm, while the size of CH is much larger than Friedel’s salt, and the size of C_3_S is less than 1 μm. From the EDS analysis results of the hexagonal flake, the energy spectrum is mainly composed of Ca, Al, O, and Cl, which is consistent with the main components of Friedel’s salt. Therefore, the spot point with a size of 2.19 μm to 2.80 μm on SEM is identified to be Friedel’s salt.

It can be seen in Figure 12a,b that CSRPC has a more dense skeleton structure than QSRPC. This is because CS calcined at a high temperature contains a large number of active glass beads, which is conducive to water reduction, densification, and homogenization, and can also play the role of micro-aggregate. In addition, CS contains a large amount of active SiO_2_ and Al_2_O_3_, which can react with the cement hydration product CH to generate cementitious substances, enhancing the compactness and strength of RPC.

It can be seen in Figure 12a,c,d that the CSRPC before freeze–thaw has a denser skeleton structure after chloride ion diffusion. This is because when chloride ions diffuse into RPC, part of them will be absorbed and solidified by C-S-H gel, and the other will react with its internal aluminum phase hydrates to generate Friedel’s salt, with a larger volume than the hydration reactants, which can fill small pores and refine large pores, reducing the impact of freeze–thaw damage on CSRPC.

#### 3.4.3. Effect of Freeze–Thaw Cycles on Chloride Diffusion of CSRPC

Figure 13 and Figure 14 show the XRD and DTG analysis results before and after 250 freeze–thaw cycles. Table 9 shows the chemical composition and content analysis results of QSRPC and CSRPC before and after 250 freeze–thaw cycles.

As can be seen in Figure 13, the main components of CSRPC and QSRPC are Friedel’s salt, CH, and SiO_2_. As can be seen in Figure 14, the first to the fifth peaks of the DTG curves are located at 20~100 °C, 100~200 °C, 310~380 °C, 400~450 °C, and 700~800 °C, respectively, which are caused by the evaporation of free water, C-S-H dehydroxylation, the decomposition of Friedel’s salt, the decomposition of CH, and the decarburization of calcium silicate, respectively.

It can be seen in Table 9 that the content of Friedel’s salt and CH in QSRPC and CSRPC after 250 freeze–thaw cycles is higher than before freeze–thaw cycles, while the content of CaSiO_3_ after 250 freeze–thaw cycles is less than the freeze–thaw cycles. The content of Friedel’s salt of CSRPC is less than QSRPC, whether before or after the freeze and thaw cycles. The crack caused by the expansion of Friedel’s salt in water has less influence on CSRPC.

It can be seen in Figure 15 that the pores where chloride ions diffuse in RPC further expand under the influence of the frost-heaving force, enhancing the diffusion capacity of chloride ions, but CSRPC has better resistance to chloride ion diffusion than QSRPC. This is because CS contains more fine particles, which play the role of micro-aggregates in the RPC skeleton and effectively block chloride ion diffusion channels between coarse particles. In addition, the active silica and aluminum oxide contained in CS can react with CH generated by the hydration reaction of cement to form C-S-H gel, which increases the compactness of RPC and improves the resistance of RPC to chloride ion diffusion after freeze–thaw cycles.

## 4. Conclusions

In this paper, CS is used to replace QS to improve the chloride diffusion resistance of RPC after freeze–thaw damage, and the effect of the CS substitution rate on the 28d UCS and bulk density of CSRPC is studied. In order to reveal the law of chloride ion diffusion under freeze–thaw damage, the influence of freeze–thaw cycles on the pore size distribution of CSRPC is studied by the NMR method, and the change in electrical flux after freezing and thawing cycles was studied and verified using electrical flux tests. The mathematical relationship between the chloride diffusion coefficient and pore fractal characteristic parameters is established. The chloride diffusion resistance mechanism of CSRPC after freeze–thaw cycles is studied by the SEM/EDS, XRD, and TG methods from the perspective of micro-pore structures and chemical products. The following main conclusions are drawn:(1)When the substitution rate of CS reaches 70%, the 28d UCS of CSRPC reaches the maximum; when it is lower than this substitution rate, CS shows pozzolanic activity, generating a small amount of C-S-H gel to cement the aggregate; and when it is greater than this substitution rate, the C-S-H gel reacts with water to cause the volume to expand, causing RPC to crack. At the same freeze–thaw cycles, the 28d UCS of CSRPC with a 100% CS substitution rate is higher than QSRPC, indicating that CSRPC has better freeze–thaw damage resistance than QSRPC.(2)With the increase in freeze–thaw cycles, the proportion of micro-pores gradually decreases, while the proportion of meso-pores and macro-pores gradually increases. The substitution of CS for QS in RPC can effectively reduce the proportion of meso-pores and macro-pores and alleviate the degree of freeze–thaw damage of RPC. The active SiO_2_ in CS has pozzolanic activity, and the reaction with CH will lead to an increase in the curvature of the pore throat, a reduction in the connectivity of the pore, and an increase in chloride ion diffusion resistance.(3)The fractal dimensions of different sizes of pores are determined, and the pore fractal characteristic parameter T is used to characterize the pore complexity and resistance to the chloride ion penetration of CSRPC. Under freeze–thaw cycles, the chloride ion diffusion coefficient of CSRPC with different CS substitution rates has a linear relationship with the pore fractal dimension.(4)When chloride ion diffuses into RPC, one part is absorbed and solidified by C-S-H gel, and the other part reacts with aluminum hydrate to generate Friedel’s salt, whose volume will expand when encountering water, filling small pores, refining large pores, and increasing the compactness of RPC after freeze–thaw damage. The active silica in CS reacts with the cement hydration product CH to form C-S-H gel, enhancing the chloride ion diffusion resistance of CSRPC after freeze–thaw cycles.

## Figures and Tables

**Figure 1 materials-17-00212-f001:**
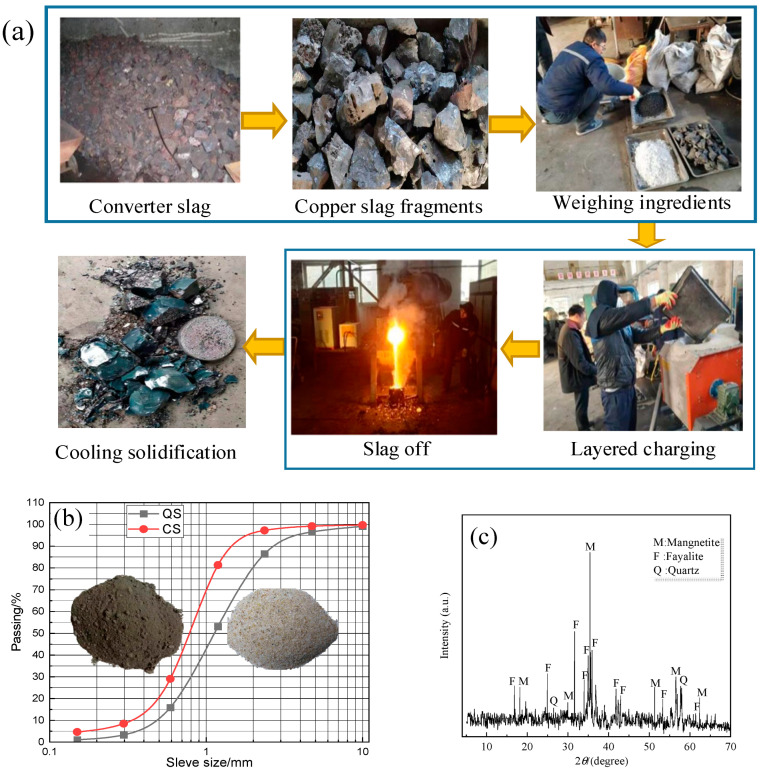
The production process and the analysis of grading and the XRD of CS: (**a**) The production process of CS (**b**) The analysis of grading of CS (**c**) The XRD of CS.

**Figure 2 materials-17-00212-f002:**
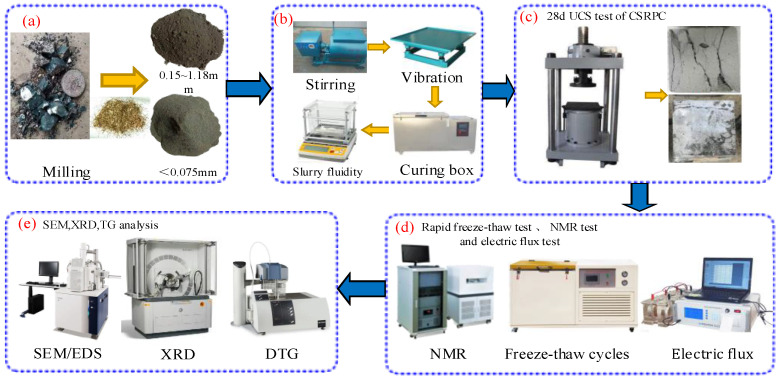
Test scheme and process: (**a**) The grinding process of CS (**b**) The manufacturing process of CSRPC (**c**) 28d UCS test of CSRPC (**d**) Rapid freeze-thaw test, NMR test and electric flux test (**e**) SEM, XRD, TG analysis.

**Figure 3 materials-17-00212-f003:**
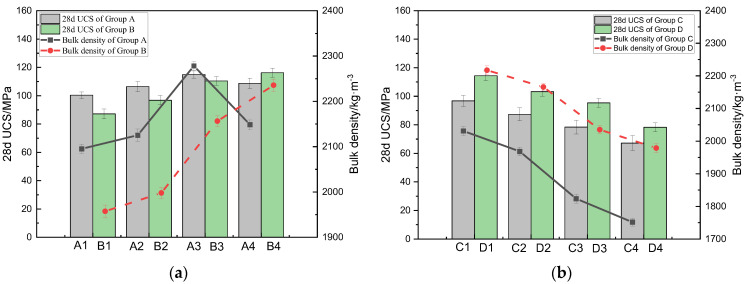
The 28d UCS and bulk density of CSRPC with different CS substitution rates: (**a**) Comparison group A and B (**b**) Comparison group C and D.

**Figure 4 materials-17-00212-f004:**
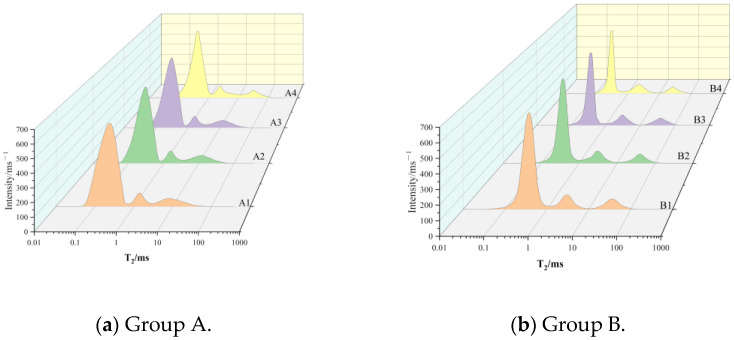
T_2_ spectrum of CSRPC.

**Figure 5 materials-17-00212-f005:**
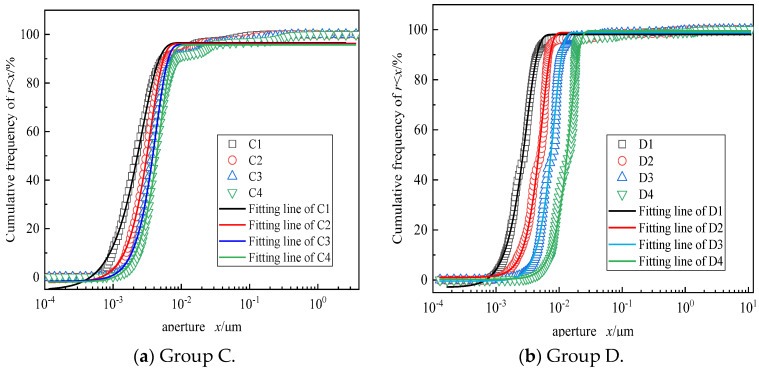
Cumulative probability density curve of pore size distribution.

**Figure 6 materials-17-00212-f006:**
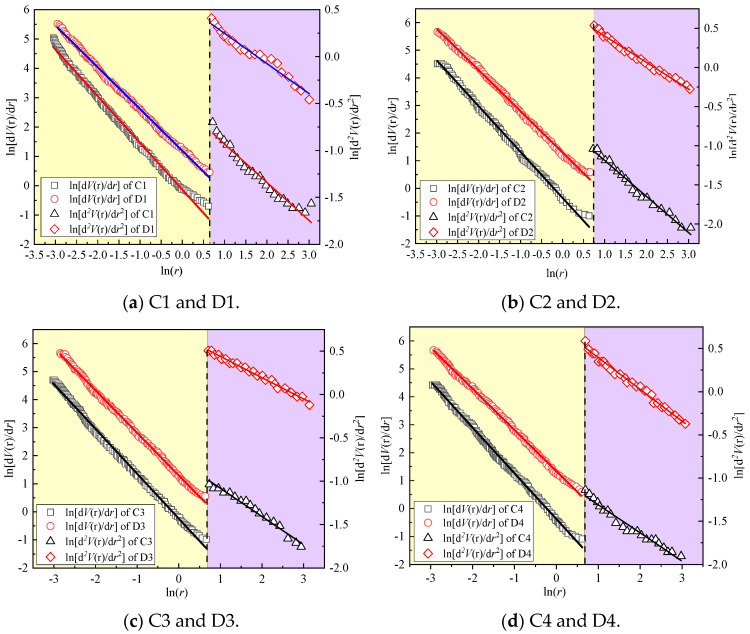
Fractal dimension of the pore throat and surface.

**Figure 7 materials-17-00212-f007:**
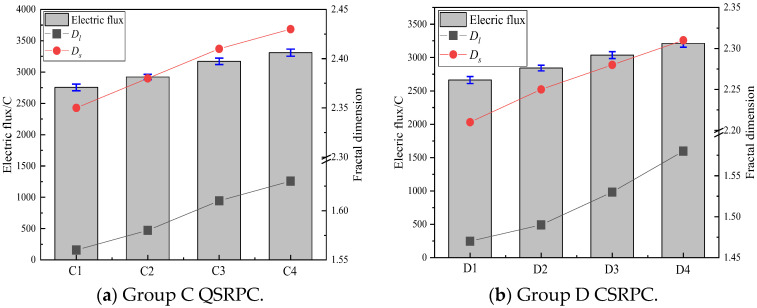
The electric flux and fractal dimension of groups C and D.

**Figure 8 materials-17-00212-f008:**
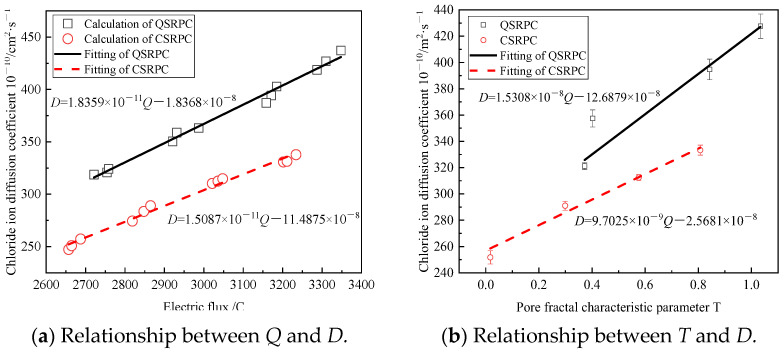
Relationship between the electric flux and pore structure characteristic parameter *T*.

**Figure 9 materials-17-00212-f009:**
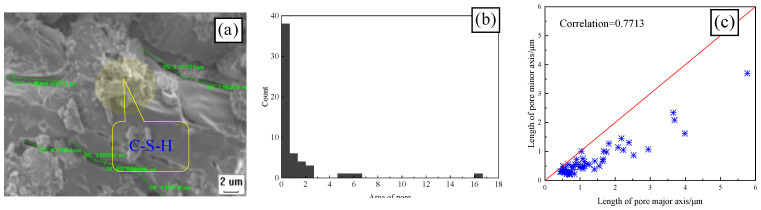
CSRPC with a CS substitution rate = 0% (**a**) SEM; (**b**) pore area; (**c**) the correlation of the length of the major and minor axes.

**Figure 10 materials-17-00212-f010:**
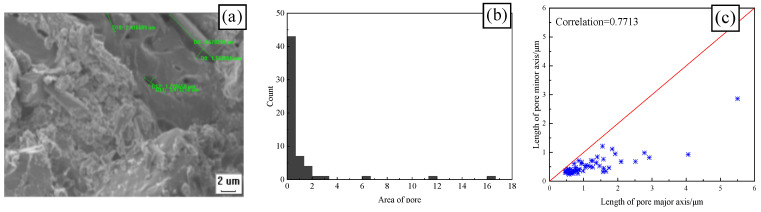
CSRPC with a CS substitution rate = 100% (**a**) SEM; (**b**) pore area; (**c**) the correlation of the length of the major and minor axes.

**Figure 11 materials-17-00212-f011:**
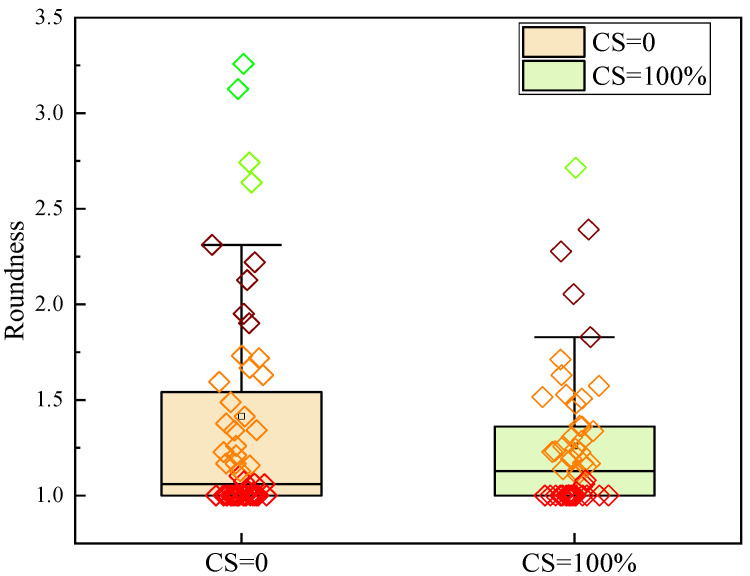
Pore roundness of CSRPC with CS substitution rates of 0 and 100%.

**Figure 12 materials-17-00212-f012:**
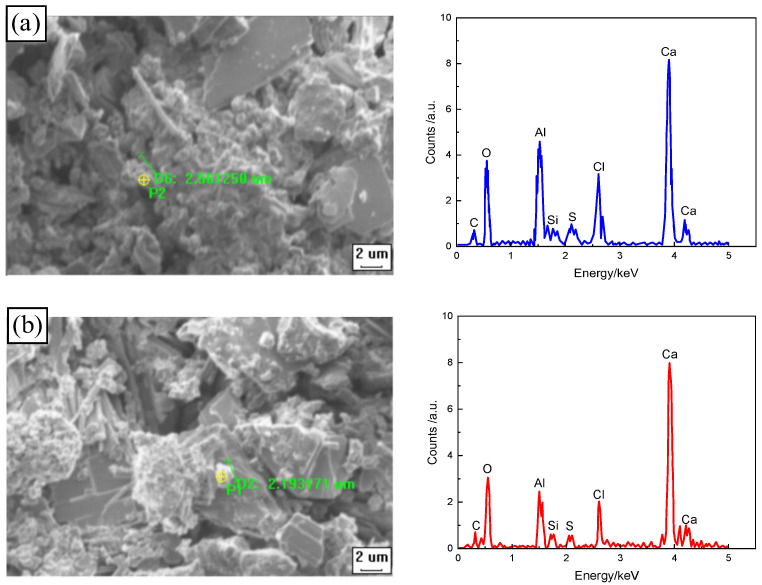
SEM/EDS image of the effect of freeze–thaw cycles on chloride ion penetration of CSRPC: (**a**) after 250 CSRPC freeze–thaw cycles, chloride ion diffusion occurs; (**b**) after 250 QSRPC freeze–thaw cycles, chloride ion diffusion occurs; (**c**) before CSRPC freeze–thaw cycles, chloride ion diffusion occurs; (**d**) after 250 CSRPC freeze–thaw cycles, there is no chloride ion diffusion.

**Figure 13 materials-17-00212-f013:**
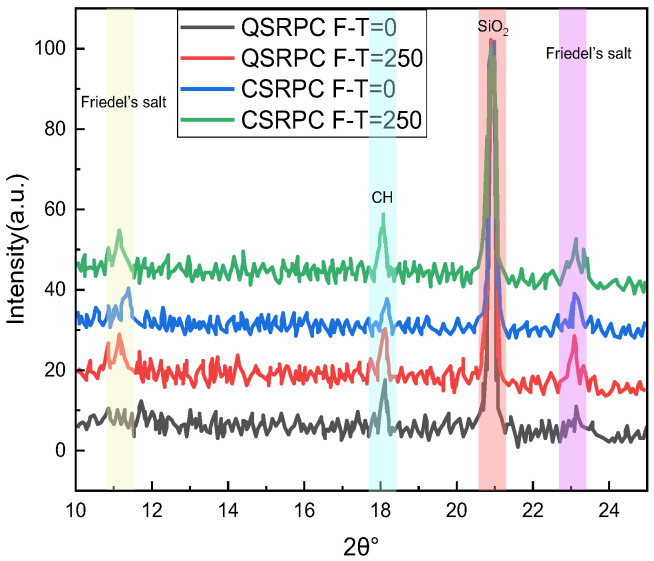
XRD analysis.

**Figure 14 materials-17-00212-f014:**
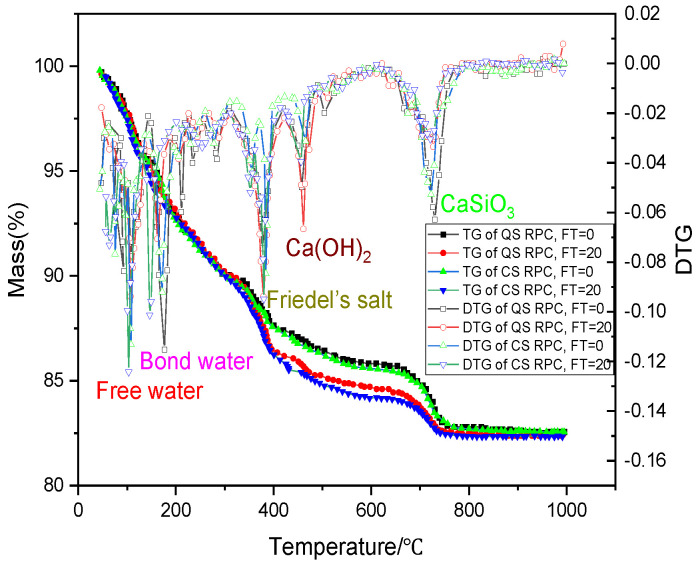
DTG analysis.

**Figure 15 materials-17-00212-f015:**
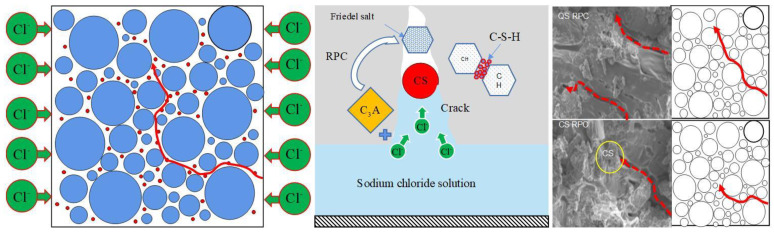
Chloride ion penetration mechanism.

**Table 1 materials-17-00212-t001:** Basic physical parameters of copper slag.

Physical Parameters	Loose Unit Weight/g	Density/g·cm^−3^	Mohs Hardness Degrees	Large Fragment Diameter/mm
Results	2.0~2.4	3.5~4.3	7~9	4.25~20

**Table 2 materials-17-00212-t002:** Chemical composition of CS.

Chemical Composition	Fe_2_O_3_	CuO	CaO	SiO_2_	Al_2_O_3_	Mn_2_O_3_	Na_2_O	Others
Content/%	18.29	1.2	32.5	29.11	11.20	0.42	0.58	6.7

**Table 3 materials-17-00212-t003:** Chemical composition and physical properties of cement.

Chemical Composition/%						Cement Burning Loss/%	Fineness/%
CaO	SiO_2_	Al_2_O_3_	Fe_2_O_3_	MgO	SO_3_		
64.37	23.27	3.85	4.57	2.37	≤2.5	≤3.5	≤6

**Table 4 materials-17-00212-t004:** Particle size analysis of silica fume.

Particle Size Distribution				Median Diameter/μm	Specific Surface Area/m^2^·g^−1^
Grain Size/μm	<2.6	<1.6	<1.3		
Proportion/%	31	57	12	1.6	14.3

**Table 5 materials-17-00212-t005:** Chemical composition of silica fume.

Chemical Composition	SiO_2_	C	CaO	Fe_2_O_3_	MgO	Na_2_O	K_2_O	Silica Fume Burning Loss
Content/%	92.95	2.06	0.95	0.49	0.37	0.16	0.87	2.2

**Table 6 materials-17-00212-t006:** Experimental scheme design.

NO.	CS Substitution Rate	Freeze–Thaw Cycles	Freeze–Thaw Condition
A1	0	0	-
A2	50	0	
A3	70	0	
A4	100	0	
B1	0	50	Freezing temperature = −20 °C, Thawing temperature = 20 °C
B2	50	50	
B3	70	50	
B4	100	50	
C1	0	100	
C2	0	150	
C3	0	200	
C4	0	250	
D1	100	100	
D2	100	150	
D3	100	200	
D4	100	250	

**Table 7 materials-17-00212-t007:** Statistics of the spectral area and pore ratio.

NO.	Spectral Area			Total Spectral Area	Spectral Area Growth Rate/%	Proportion		
	Micro-Pore	Meso-Pore	Macro-Pore			Micro-Pore	Meso-Pore	Macro-Pore
A1	1864.25	672.15	7.21	2622.61	-	71.08	25.63	3.29
A2	1635.21	623.58	6.86	2334.65	−10.98	70.04	26.71	3.25
A3	1513.58	601.87	6.14	2178.59	−16.93	69.48	27.63	2.90
A4	1378.39	587.35	5.87	2018.61	−23.03	68.28	29.10	2.62
B1	1935.49	688.27	8.69	2715.45	-	71.28	25.35	3.38
B2	1727.15	665.89	7.25	2480.29	−8.66	69.64	26.85	3.52
B3	1686.75	633.48	6.28	2396.51	−11.75	70.38	26.43	3.18
B4	1536.78	592.15	6.12	2195.05	−19.16	70.01	26.98	3.01
C1	2257.64	757.21	10.35	3125.2	-	72.24	24.23	3.53
C2	2536.25	925.43	23.87	3585.55	14.73	70.74	25.81	3.45
C3	2775.38	1238.79	56.21	4170.38	33.44	66.55	29.70	3.75
C4	3162.41	1765.21	77.15	5104.77	47.47	61.95	34.58	3.47
D1	1931.47	535.42	4.15	2541.04	-	76.01	23.07	2.92
D2	2127.36	713.45	15.47	2936.28	15.55	72.45	24.30	3.25
D3	2298.75	1075.27	22.36	3496.38	37.60	65.75	30.75	3.50
D4	2871.43	1445.86	59.32	4476.61	76.17	64.14	32.30	3.56

**Table 8 materials-17-00212-t008:** The fractal dimension and fractal characteristic parameter.

NO.	$D_{l}$	$D_{s}$	φ/%	*X* × 10^−3^/μm	*n*	$A_{t}$	T
C1	1.56	2.35	3.25	2.63	1.69	1509.27	0.3724
C2	1.58	2.38	3.87	3.53	2.28	1621.37	0.4025
C3	1.61	2.41	4.36	4.27	2.50	1725.35	0.8425
C4	1.63	2.43	4.78	4.85	2.52	1921.28	1.0341
D1	1.47	2.21	2.71	3.04	2.37	1623.84	0.0174
D2	1.49	2.25	2.97	5.35	2.95	1735.96	0.2990
D3	1.53	2.28	3.12	8.29	3.04	1921.08	0.5758
D4	1.58	2.31	3.47	15.09	3.94	1978.85	0.8075

**Table 9 materials-17-00212-t009:** The content of the chemical composition of RPC before and after 250 freeze–thaw cycles.

RPC	Freeze–Thaw Cycles	Free Water (%)	Bond Water (%)	Friedel’s Salt (%)	Ca(OH)_2_ (%)	CaSiO_3_ (%)
QSRPC	0	10.35	22.36	4.87	12.43	19.26
QSRPC	250	12.17	25.48	8.35	15.46	18.51
CSRPC	0	11.57	18.24	3.27	8.37	14.35
CSRPC	250	8.75	16.78	6.68	10.25	8.92

## Data Availability

Data are contained within the article.

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
