# Peer review of "A Study on Chloride Corrosion Resistance of Reactive Powder Concrete (RPC) with Copper Slag Replacing Quartz Sand under Freeze–Thaw Conditions"

_materials, 2023, doi:10.3390/ma17010212_

Round 1

Reviewer 1 Report

Comments and Suggestions for Authors

Please find an attached!

Reviewer 2 Report

Comments and Suggestions for Authors

The paper provides valuable insights into the potential benefits of using copper slag in RPC, there are some areas that could be improved upon to enhance the clarity and rigor of the study.

1. The English language used throughout the paper can be improved to increase readability. For example, the overuse of phrases like "as can be seen in Fig" can be replaced.

2. The authors should double-check their use of chemical notions, such as C3S. Ensuring accurate nomenclature.

3. It would be beneficial for the author to provide the linear equation in Fig. 9c and Fig. 10C and discuss more the meaning of the coefficients.

4. The addition of an estimation for the diffusion coefficient of Cl- in both cases in Fig. 15c would be highly informative. Demonstrating the benefit of CS substitution in terms of chloride ion penetration could significantly strengthen the paper's argument for the use of copper slag in RPC.

Comments on the Quality of English Language

Too many repetition in the text, such as "...as can be seen in Fig .."

Reviewer 3 Report

Comments and Suggestions for Authors

The  paper tackles very important issue from environment protection and resources saving point of view - namely replacement of quartz sand with copper slag in Reactive Powder Concrete. Considerable amount of work is carried out to study the influence of this replacement on the mechanical and chemical properties of the concrete, as well as its resistance to chloride ion corrosion and the influence of freeze-thaw cycles on the corrosion resistance.

Presented data are of interest to engineers working the area of metallurgical waste utilization.

However, before to be published the text needs careful consideration and corrections.

In order to facilitate this process I am attaching the paper file with comments and proposal inside.

Kind regards,

Comments on the Quality of English Language

Moderate editing of English language is needed.
